# Towards artificial general intelligence via a multimodal foundation model

Nanyi Fei[1,2,3], Zhiwu Lu [1,2✉], Yizhao Gao[1,2], Guoxing Yang[1,2], Yuqi Huo[2,3], Jingyuan Wen[1,2], Haoyu Lu[1,2], Ruihua Song[1,2], Xin Gao [4], Tao Xiang[5], Hao Sun [1,2✉] & Ji-Rong Wen [1,2,3✉]

The fundamental goal of artificial intelligence (AI) is to mimic the core cognitive activities of human. Despite tremendous success in the AI research, most of existing methods have only single-cognitive ability. To overcome this limitation and take a solid step towards artificial general intelligence (AGI), we develop a foundation model pre-trained with huge multimodal data, which can be quickly adapted for various downstream cognitive tasks. To achieve this goal, we propose to pre-train our foundation model by self-supervised learning with weak semantic correlation data crawled from the Internet and show that promising results can be obtained on a wide range of downstream tasks. Particularly, with the developed model-interpretability tools, we demonstrate that strong imagination ability is now possessed by our foundation model. We believe that our work makes a transformative stride towards AGI, from our common practice of "weak or narrow AI" to that of "strong or generalized AI".

[1] Gaoling School of Artificial Intelligence, Renmin University of China, Beijing, China. [2] Beijing Key Laboratory of Big Data Management and Analysis Methods, Beijing, China. [3] School of Information, Renmin University of China, Beijing, China. [4] Computer, Electrical and Mathematical Sciences and Engineering Division, King Abdullah University of Science and Technology, Thuwal, Saudi Arabia. [5] Department of Electrical and Electronic Engineering, University of Surrey, Guildford, UK. ✉email: luzhiwu@ruc.edu.cn; haosun@ruc.edu.cn; jrwen@ruc.edu.cn

Science fictions and sci-fi films, that describe highly intelligent computer minds, robots, or even human-shaped ones, can be said to understand or have primitive cognitive abilities analogous to human intelligence. Since this form of human-level artificial intelligence (AI) is too far from reality hitherto, researchers change to set a less ambitious goal of achieving artificial general intelligence (AGI). Despite not being precisely defined, AGI is broadly agreed to have several key features[1] including: (1) matching or exceeding human performance across a broad class of cognitive tasks (e.g., perception, reading comprehension, and reasoning) in a variety of contexts and environments; (2) possessing the ability to handle problems quite different from those anticipated by its creators; and (3) being able to generalize/transfer the learned knowledge from one context to others. As we can imagine, devising and obtaining an AGI system would not only accelerate the AI research itself, but also benefit a wide range of AI+ fields including neuroscience, healthcare, and biomedicine.

In recent years, deep learning[2] has achieved tremendous successes in various AI research areas such as computer vision (CV) and natural language processing (NLP). For example, deep residual networks (ResNets)[3] have already surpassed human performance on image classification. The language model RoBERTa[4] has also outperformed human on several natural language understanding tasks of the GLUE benchmark[5]. Relation networks[6] devised by DeepMind have achieved super-human performance on a relational reasoning dataset. However, most of existing AI advances only focus on approaching or exceeding human intelligence on single cognitive ability (e.g., image classification, language understanding, or relational reasoning). To overcome such a limitation and take a solid step to AGI, we develop a foundation model pre-trained with huge multimodal (visual and textual) data such that it can be quickly adapted for a broad class of downstream cognitive tasks.

Our motivations are two-fold: (1) Foundation models[7] (also well-known as pre-trained models) are established because they are exactly designed to be adapted (e.g., finetuned) to various downstream cognitive tasks by pre-training on broad data at scale. Importantly, foundation models are closely related to two breakthroughs of MIT Technology Review's "10 Breakthrough Technologies 2021"[8]: GPT-3[9] (a pre-trained language model) and multi-skilled AI. (2) Our choice of learning from huge multimodal data is inspired by the fact that most human intelligent behaviors are exhibited in a multimodal context using visual-textual content as the primary carrier of knowledge and means of communication (see Fig. 1a). Indeed, researchers have reported that a subset of neurons in the human medial temporal lobe can be selectively activated by representations of a specific object/scene across different sensory modalities (e.g., pictures, written names, and spoken names)[10,11]. Although the mechanism of cross-modal alignment in our brain is unknown, this still suggests that human brain neurons are able to process multimodal information and encode concepts into invariant representations. Overall, we believe that pre-training a large-scale multimodal foundation model is indeed a potential approach to achieving AGI.

Multimodal (visual and textual) foundation models[12,13] typically take image-text pairs as input and model the correlation between two different modalities in their pre-training data. Although existing multimodal foundation models have shown promising results on fast learning/transfer and cross-modal understanding tasks, the majority of them[12,14–18] make the assumption of strong semantic correlation over the input image-text pairs (e.g., image-caption pairs) and expect exact matches between the objects/regions in an image and the words in a piece of text (see Fig. 1b). This seriously limits these models'

generalization abilities because the strong semantic correlation assumption is often invalid in the real world and multimodal data following this assumption is limited (e.g., only millions of image-caption pairs are collected by years of human annotation). This situation becomes worse when latest multimodal foundation models[12,17,19–21] often employ object detectors to obtain meaningful image regions and adopt a single-tower network architecture for better modeling the fine-grained region-word matching (i.e., taking the concatenation of image regions and text words as input). These two common practices (i.e., object detectors and the single-tower architecture) are both computationally costly and thus unsuited for real-world applications. Particularly, as for the latter, given a query in cross-modal retrieval (text-to-image or image-to-text), all possible query-candidate pairs need to be fed into the model to compute matching scores, resulting in large latency in retrieval.

To address the above issues, we develop a large-scale multimodal foundation model dubbed Bridging-Vision-and-Language (BriVL) by self-supervised learning[22–25] from huge multimodal data. Firstly, to build our pre-training data collection, we choose to exploit weak semantic correlation data (see Fig. 1b) available on the Internet without any need of human annotation (i.e., we crawl a total of 650 million image-text pairs from the web). Importantly, such huge weak semantic correlation data contains complicated/abstract human emotions and thoughts. Therefore, comparing to modeling strong semantic correlation data by direct image-to-text "translation" in previous works[12,17,19–21], modeling weak semantic correlation data by image-text matching would help us obtain a more cognitive model. Secondly, to design our network architecture, since there do not necessarily exist fine-grained region-word matches between image and text modalities, we drop the time-consuming object detectors and adopt a simple two-tower architecture (instead of the single-tower one), which encodes image and text inputs using two separate encoders (see Fig. 1a). Note that the two-tower architecture has a clear advantage in efficiency during inference, as the embeddings of candidates can be computed and indexed before querying, meeting the latency requirement of real-world applications. Thirdly, with the advancement of large-scale distributed training techniques[26,27] and self-supervised learning[22–25], learning from huge unannotated multimodal data becomes possible. Specifically, to model the weak image-text correlation and learn a unified semantic space where global-level image/text embeddings are aligned, we devise a cross-modal contrastive learning (CL) algorithm, where CL is a special form of self-supervised learning that is initially developed in single-modality (i.e., images)[28–31] with the learning objective of keeping the positive samples close and pushing away the negative ones. The proposed network and algorithm designs are detailed in Methods.

Although OpenAI CLIP[13] and Google ALIGN[32] are most closely related to our BriVL, there exist two main differences between BriVL and these two latest models: (1) We follow the weak semantic correlation assumption and construct a huge dataset crawled from the Internet, and only pornographic/sensitive data are filtered out in our collected dataset. In contrast, CLIP only keeps image-text pairs with high word frequency (i.e., the long-tail concepts are discarded), while ALIGN also filters its pre-training dataset by some rules (e.g., excluding texts shared by more than 10 images, excluding texts with extremely low word frequency, and excluding texts that are too long or too short). Our dataset thus preserves a data distribution closer to that of the real world. (2) Inspired by the single-modal contrastive learning (CL) algorithm MoCo[29], our BriVL model adopts a momentum mechanism to dynamically maintain queues of negative samples across different training batches. In this way, we have a large negative sample size (crucial for CL) while using a relatively small

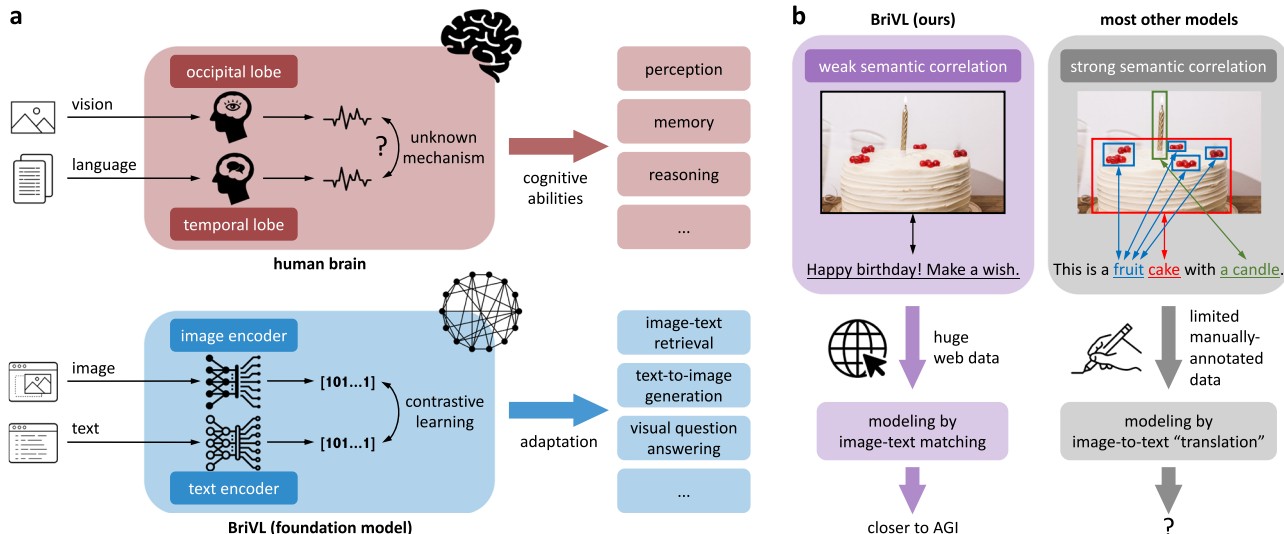

**Fig. 1 Overarching concept of our BriVL model with weak training data assumption. a** Comparison between the human brain and our multimodal foundation model BriVL (Bridging-Vision-and-Language) for coping with both vision and language information. **b** Comparison between modeling weak semantic correlation data and modeling strong semantic correlation data.

batch size (to reduce GPU memory footprint). On the contrary, both CLIP and ALIGN use negative samples within each training batch, requiring a large batch size (i.e., a great demand for GPU memories/resources). More technical differences can be found in Methods.

We conduct extensive experiments on various downstream cognitive tasks (e.g., news classification in single-modal and visual question answering in cross-modal) and show that our foundation model BriVL achieves promising results, demonstrating its cross-modal understanding ability and cross-domain learning/transfer ability. Although our BriVL is only pre-trained with an image-text matching learning objective, its strong generalization ability has already satisfied some of the key features that an AGI system should have. Importantly, with a couple of model-interpretability tools developed in this work, we manage to visually reveal how a multimodal foundation model reasonably and logically imagines when words or sentences are told, showing that our BriVL exhibits strong imagination ability. A closer examination reveals that the possession of strong imagination is mainly due to the fact that our BriVL leverages weak semantic correlation data in large-scale multimodal pre-training. Overall, these findings indicate that pre-training a multimodal (visual and textual) foundation model can make a giant stride towards AGI. With more sensory modalities exploited for multimodal pre-training and further exploration on more advancing foundation models, we believe that we are approaching AGI and our work will have a broad impact on a variety of AI+ fields including neuroscience, healthcare, and biomedicine.

## Results

Our BriVL model has been pre-trained based on a huge weak semantic correlation dataset collected from public web sources. The resulting model possesses excellent imagination ability, evidenced by Neural Network Visualization, Text-to-Image Generation and multiple downstream tasks (i.e., remote sensing scene classification, news classification, cross-modal retrieval, and visual question answering), which are discussed in detail in this section.

**Pre-training data collection**. We construct a huge web-crawled multi-source image-text dataset called weak semantic correlation dataset (WSCD) as our pre-training data collection. WSCD

collects Chinese image-text pairs from multiple sources on the web, including news, encyclopedia, and social media. Concretely, images from these data sources, together with their corresponding/surrounding text descriptions, are used to form image-text pairs. Since the obtained image-text pairs are crawled from the web, the image and the text of each pair are expected to be weakly correlated. For example, an image from social media that contains people having a good time with friends tends to have a simple title of "What a nice day!", without any finer-grained description of the image content. Note that we only filter out the pornographic/sensitive data from WSCD, without any form of editing/modification to the raw data to preserve the natural data distribution. Totally, WSCD has around 650 million image-text pairs covering a wide range of topics such as sports, lifestyle, and movie posters. Since WSCD is based on Chinese, English texts of all experiments in this section are translated into Chinese for our BriVL. Furthermore, we pre-train our BriVL on an English dataset and show results on English tasks in Supplementary Note Fig. S3, indicating that our foundation model also provides a feasible solution closer to AGI beyond specific languages.

**Neural network visualization**. Humans have the ability (or even instinct) that scenes, e.g., in the context of images, come into our minds when we hear words or descriptive sentences. As for our BriVL, once pre-trained on such a vast amount of loosely aligned image-text pairs (see Methods), we are fascinated by what exactly it would imagine when texts are given. Instead of examining it indirectly through downstream tasks, we extend Feature Visualization (FeaVis)[33] to see the visual responses (i.e., imagination) of BriVL to semantic inputs directly. FeaVis is an algorithm designed only to visualize the features of convolutional neural networks (CNNs). However, given a large-scale cross-modal foundation model like our BriVL, we can visualize any text input by using the joint image-text embedding space as the bridge. Concretely, we first input a piece of text and obtain its text embedding through the text encoder of BriVL. Next, we randomly initialize a noisy image and also get an image embedding through the image encoder. Since the input image is randomly initialized, its embedding does not match that of the input text. We thus define the objective of matching the two embeddings and back-propagate the resultant gradients to update the input image. Note that we do not use any additional module or data for

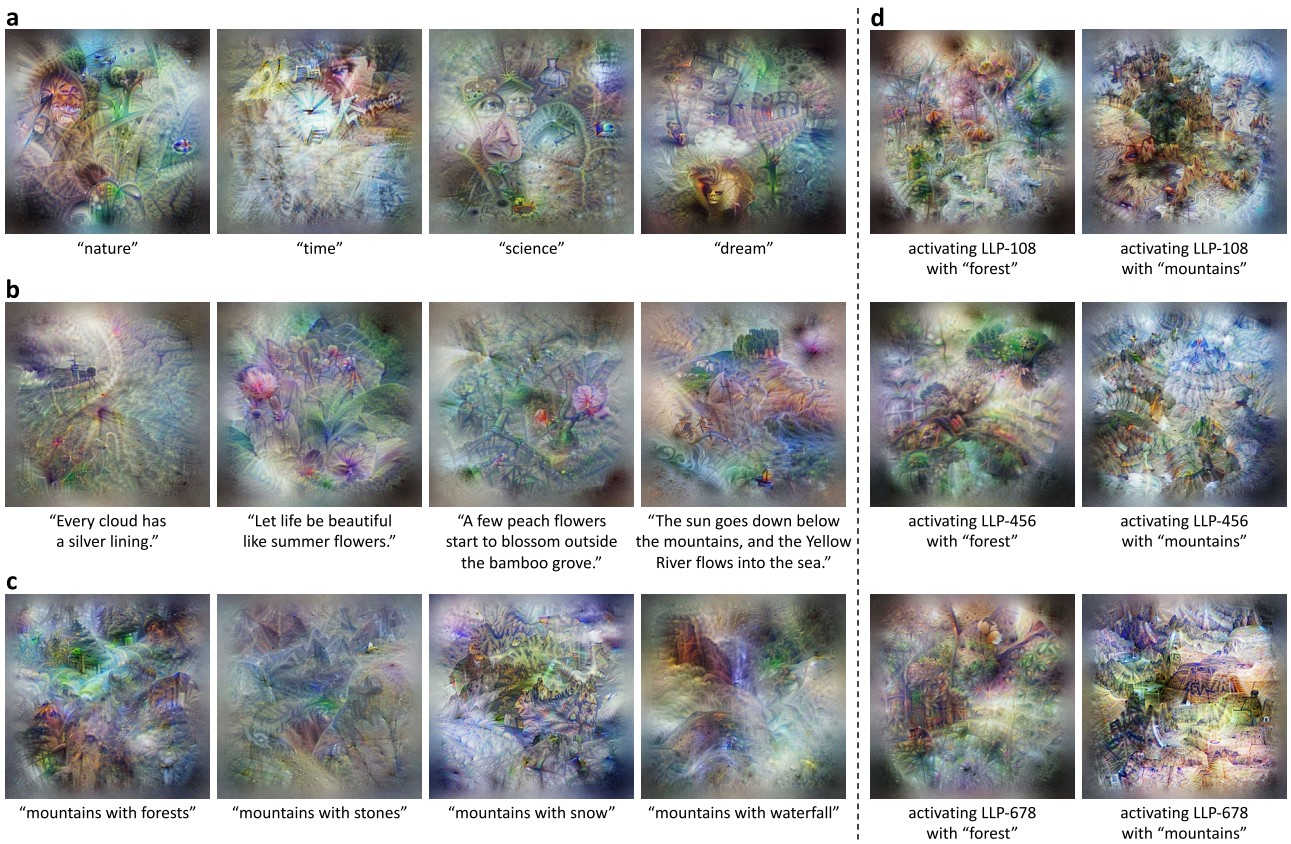

**Fig. 2 Network and neuron visualizations of our BriVL's imagination. a** Visualizations of the final embedding layer of BriVL w.r.t. high-level concepts. **b** Visualizations of the final embedding layer of BriVL w.r.t. free-form text inputs. **c** Visualizations of the final embedding layer of BriVL with semantic restrictions related to "mountains with". **d** Visualizations for different neurons of BriVL with semantic restrictions "forest" and "mountains".

visualization, while the pre-trained BriVL is frozen during the whole process. The finally obtained image thus depicts a clear picture of what BriVL imagines about the input text. The visualizations of different semantic inputs are shown in Fig. 2. Note that the input texts are originally in Chinese and translated into English for illustration purpose.

We first present the imagination ability of BriVL to high-level concepts in Fig. 2a. It can be seen that, even though these concepts are rather abstract, the visualizations are able to show concrete embodiment of these concepts (e.g., "nature": plants like grass; "time": a clock; "science": a face with glasses and a conical flask; "dream": cloud, a bridge leading to a door, and the dream-like atmosphere). This ability to generalize an abstract concept to a series of more concrete objects is a sign of learned common sense and an indication of the effectiveness of our multimodal pre-training using only weak semantic correlation data (which expose the model with abstract concepts).

In Fig. 2b, we show the imagination of sentences. The visualization of "Every cloud has a silver lining." not only embodies the sunlight behind dark clouds literally, but also seems to show a dangerous situation on the sea (the ship-like object and the waves on the left), expressing the implicit meaning of this sentence. In the visualization of "Let life be beautiful like summer flowers.", we can see a flower shrub. The next two text inputs describing more complicated scenes are both from ancient Chinese poems written with completely different grammar from most other texts in the dataset. It seems that BriVL also understands them well: for "A few peach flowers start to blossom outside the bamboo grove.", there are bamboos and pink flowers; for "The sun goes down below the mountains, and the Yellow

River flows into the sea.", we can see mountains with trees hiding the sunset, and a small boat on the river. Overall, we find that BriVL possesses strong capability of imagination given a complicated sentence as prompt.

In Fig. 2c, a few similar text inputs containing a shared prompt are used for network visualization. For "mountains with forests", there is more green area in the image; for "mountains with stones", the image is more rocky; for "mountains with snow", the ground turns into white/blue around the trees in the center; for "mountains with waterfall", we can see blue water falling down with even vapor visible. These imagination results indicate that our model is capable of linking specific objects with more general visual context.

We also present the neuron visualization results with semantic constraints in Fig. 2d. Concretely, in addition to the image-text matching loss described above, we select neurons (i.e., channels) in the feature map of the last layer before the pooling layer (LLP, short for "Last Layer before Pooling") in our image encoder and maximize the value of each neuron. Since each text input may contain many semantic contents, we can see what it is equivalent to activating one neuron under certain semantic constraint. Three neurons LLP-108, LLP-456, and LLP-678 (the number means the position of each channel in the feature map) are selected for neuron visualization. The two columns in Fig. 2d show the visualizations with text inputs "forest" and "mountains", respectively. We can clearly see that even with the same semantic constraint, activating different neurons leads to different imagination results, indicating that each text input has rich semantics with different aspects being captured by different neurons.

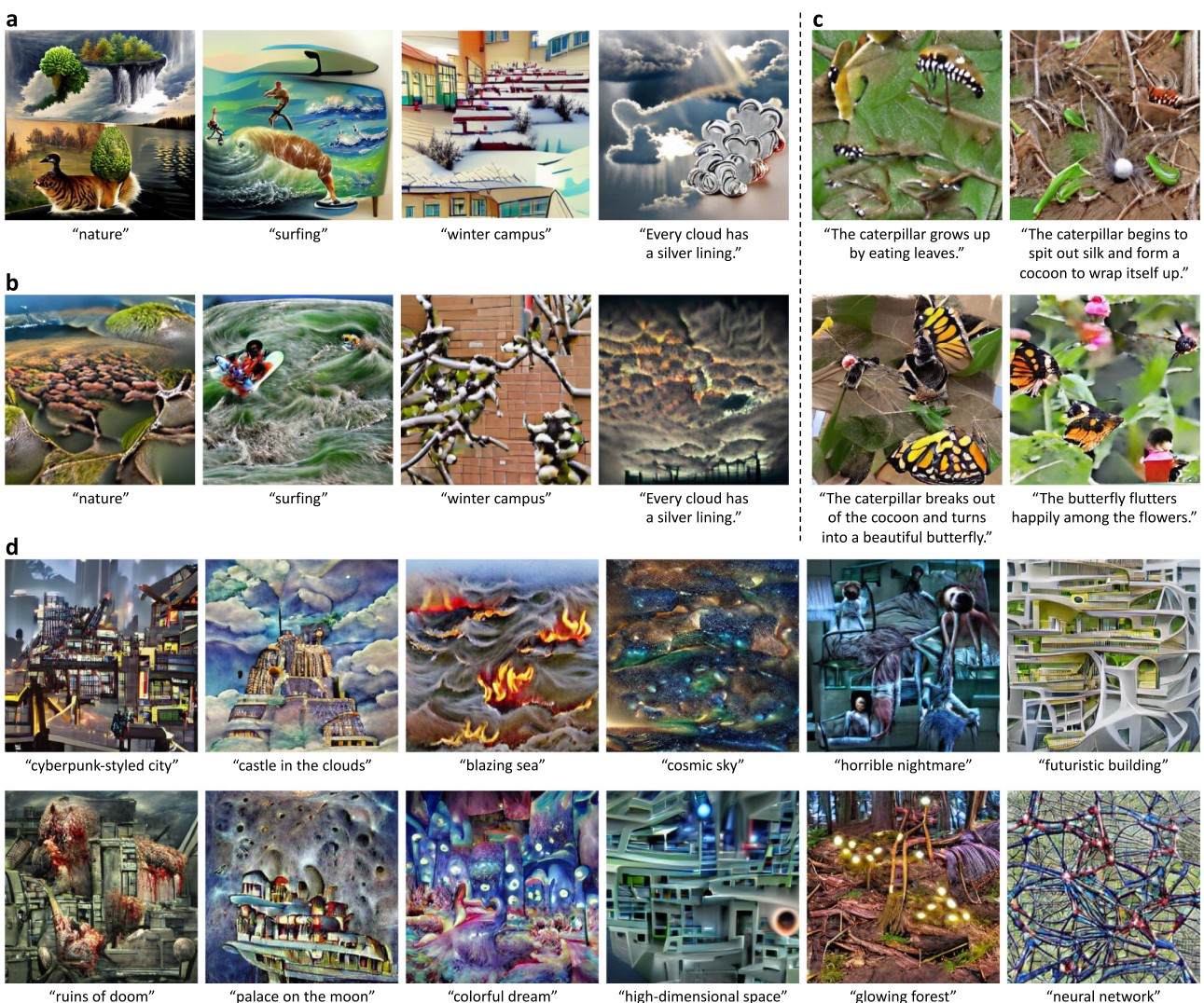

**Fig. 3 Text-to-image generation examples of clearer imagination. a** Generation examples of VQGAN inversion with CLIP (w/ ResNet-50x4).
**b** Generation examples of VQGAN inversion with our BriVL. **c** A series of generation examples by VQGAN inversion with our BriVL. **d** More generation examples by VQGAN inversion with our BriVL, where concepts/scenes are rarely seen by us humans (e.g., "blazing sea" and "glowing forest") or even do not exist in real life (e.g., "cyberpunk-styled city" and "castle in the clouds"). Note that VQGAN is pre-trained on ILSVRC-2012. BriVL, CLIP and VQGAN are all frozen during text-to-image generation.

**Text-to-image generation**. Network/neuron visualizations of the imagination are straightforward but sometimes can be hard to interpret. Here, another visualization/interpretability method is developed to make the imagined visual contents of our BriVL better understood by us human. Specifically, we utilize VQGAN[34] to generate images under the guidance of our BriVL and contrast them with those generated with CLIP[13]. A VQGAN pre-trained on the ILSVRC-2012[35] dataset is excellent in generating photo-realistic images given a sequence of tokens. Each of such token is a vector from the pre-trained token set (i.e., codebook) of VQGAN. We first randomly sample a sequence of tokens, and obtain a generated image from the pre-trained VQGAN. Next, we input the generated image into the image encoder of CLIP/BriVL and also input a piece of text into the text encoder. Finally, we define the objective of matching the image and text embeddings, and back-propagate the resultant gradients to update the initial token sequence. Like network/neuron visualization, both VQGAN and CLIP/BriVL are frozen during the generation process. The generated examples are presented in Fig. 3.

In Fig. 3a, b, we select four text inputs and show the results obtained by CLIP and our BriVL, respectively. CLIP and BriVL both understand the texts well; however, we also observe two major differences. Firstly, cartoon-styled elements tend to appear in the generated images of CLIP, while images generated by our BriVL are more real and natural. Secondly, CLIP tends to simply put elements together while BriVL-generated images are more coherent globally. The first difference may be due to the differences in the training data used by CLIP and BriVL. The images in our training data are crawled from the Internet (most are real photos), while there may be a fair amount of cartoon images in the training data of CLIP. The second difference lies in the fact that CLIP uses image-text pairs with strong semantic correlation (by word filtering) while we use weakly correlated data. This means that during multimodal pre-training, CLIP is more likely to learn the correspondence between objects (in images) and words (in texts) while BriVL is trying to understand each image with the given text as a whole.

In Fig. 3c, we consider a significantly more challenging task where a series of images should be generated according to multiple coherent sentences. Although each image in Fig. 3c is generated independently, we can observe that all four generated images are visually coherent and of the same style. This finding

**a**

| Method | Unseen/Seen Class Ratios | | | | |
|---|---|---|---|---|---|
| | 21 / 0 | 5 / 16 | 8 / 13 | 11 / 10 | 14 / 7 |
| ZSSC | / | 58.7 (0.9) | 35.4 (1.0) | 19.6 (0.5) | 15.1 (0.2) |
| CLIP w/ ResNet-50 | 50.19 | 71.98 (0.10) | 64.66 (0.09) | 59.87 (0.06) | 57.14 (0.05) |
| CLIP w/ ResNet-101 | 54.81 | 76.84 (0.11) | 70.52 (0.09) | 63.61 (0.07) | 62.01 (0.07) |
| CLIP w/ ResNet-50x4 | 56.67 | 76.02 (0.09) | 71.53 (0.07) | 64.44 (0.07) | 63.77 (0.05) |
| BriVL | **58.43** | **82.41 (0.08)** | **72.91 (0.07)** | **69.16 (0.05)** | **65.06 (0.06)** |

**b**

| Method | Unseen/Seen Class Ratios | | | | |
|---|---|---|---|---|---|
| | 30 / 0 | 8 / 22 | 12 / 18 | 16 / 14 | 20 / 10 |
| CLIP w/ ResNet-50 | 46.01 | 65.99 (0.08) | 59.15 (0.05) | 54.44 (0.05) | 51.72 (0.04) |
| CLIP w/ ResNet-101 | 48.05 | 68.71 (0.07) | 64.39 (0.06) | 57.75 (0.06) | 54.54 (0.05) |
| CLIP w/ ResNet-50x4 | 50.96 | 69.32 (0.08) | 64.30 (0.05) | 59.53 (0.06) | 56.35 (0.04) |
| BriVL | **58.12** | **76.73 (0.09)** | **71.25 (0.07)** | **67.52 (0.06)** | **64.19 (0.04)** |

**c**

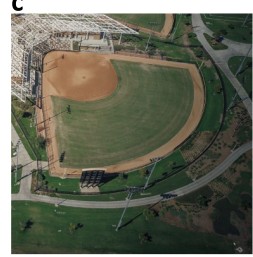

an example of the remote sensing scene "baseball field"

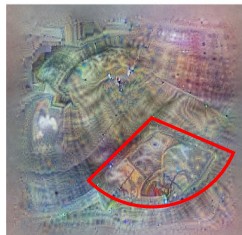

visualization of "baseball field viewed from above" with BriVL

**Fig. 4 Zero-shot remote sensing scene classification results. a** Zero-shot accuracies (%) on UCM with different unseen/seen class ratios. **b** Zero-shot accuracies (%) on AID with different unseen/seen class ratios. **c** Visualizations of "baseball field". For (**a**) and (**b**), we report standard deviations in brackets over 25 random splits. Highest results in (**a**) and (**b**) are highlighted in bold.

demonstrates another advantage of our BriVL model: although the environment and background in an image are hard to explicitly mention in the associated text, they are not neglected in our large-scale multimodal pre-training.

We present more text-to-image generation examples obtained by VQGAN inversion with our BriVL in Fig. 3d. Specifically, we choose concepts/scenes rarely seen by us humans (e.g., "blazing sea" and "glowing forest") or even those not existing in real life (e.g., "cyberpunk-styled city" and "castle in the clouds"). We find that our proposed model can generate images quite consistent with our imagination about the input concepts/scenes, indicating its strong generalization/imagination ability. This also provides evidence that the superior performance of BriVL is not due to overfitting the pre-training data since the text inputs here correspond to concepts/scenes that even do not exist in real life. In addition, these generation examples again demonstrate the advantage of pre-training BriVL on weak semantic correlation data (otherwise the fine-grained region-word matching would harm the imagination ability of BriVL).

**Remote sensing scene classification.** To show the cross-domain knowledge transfer ability and the out-of-domain imagination ability of our pre-trained BriVL, we conduct zero-shot experiments on two remote sensing scene classification benchmarks. The first dataset is UC Merced Land-Use (UCM)[36], which has 21 classes and 100 images for each class. The size of each image in UCM is 256 × 256. The second dataset is AID[37], which has 30 classes and 10,000 images in total. The size of each image in AID is 600 × 600. AID is a multi-source dataset, which makes it more challenging for scene classification than the single-source UCM. Concretely, images of each class in AID are extracted from different countries and regions around the world, and also at different times and seasons of the year under different imaging conditions. This leads to larger intra-class data diversity in AID. For each dataset, we first obtain class embeddings by inputting class names into the text encoder of CLIP/BriVL. Then for each

test image, we obtain its image embedding via the image encoder of CLIP/BriVL, and compute its cosine similarity with each class embedding to predict the class that it belongs to. Note that since the class names of these two datasets are all English, we need to translate them into Chinese to fit our BriVL (but the original class names are directly used for CLIP).

In the field of zero-shot learning (ZSL)[38], datasets typically follow the split of unseen and seen classes. Conventional ZSL models are thus trained with seen class data and evaluated on unseen class data. Although we do not need to train on seen classes, we still follow the common practice and split each dataset with different unseen/seen class ratios (the seen classes are simply not used). Under the split settings where the number of seen classes are not zero, we randomly sample 25 splits and report the standard deviations in brackets along with average accuracy.

The zero-shot classification results on UCM are shown in the table of Fig. 4a. Our BriVL is compared to a strong baseline ZSSC[39] specially designed for zero-shot remote sensing scene classification, and also CLIP with different CNN backbones. We can see that large-scale cross-modal foundation models achieve far higher rates compared with ZSSC, indicating their strong cross-domain knowledge transfer abilities. Moreover, our classification rates are also higher than those of all CLIP models with different CNNs, which is impressive considering the loss in English-to-Chinese translation and also cultural differences (CLIP is trained on English data while we use data crawled from Chinese Internet). Results on another dataset AID are shown in the table of Fig. 4b. Since we did not find methods conducting ZSL experiments on AID, we only make comparisons with CLIP variations. As we have mentioned, AID is more challenging than UCM, which is also reflected by the much worse performance of CLIP variations on AID than on UCM. However, our BriVL achieves similar performance on the two datasets when evaluated over all data, and the gap between BriVL and CLIP is larger on AID than that on UCM. This means that BriVL has stronger generalization ability and can cope with more complicated situations.

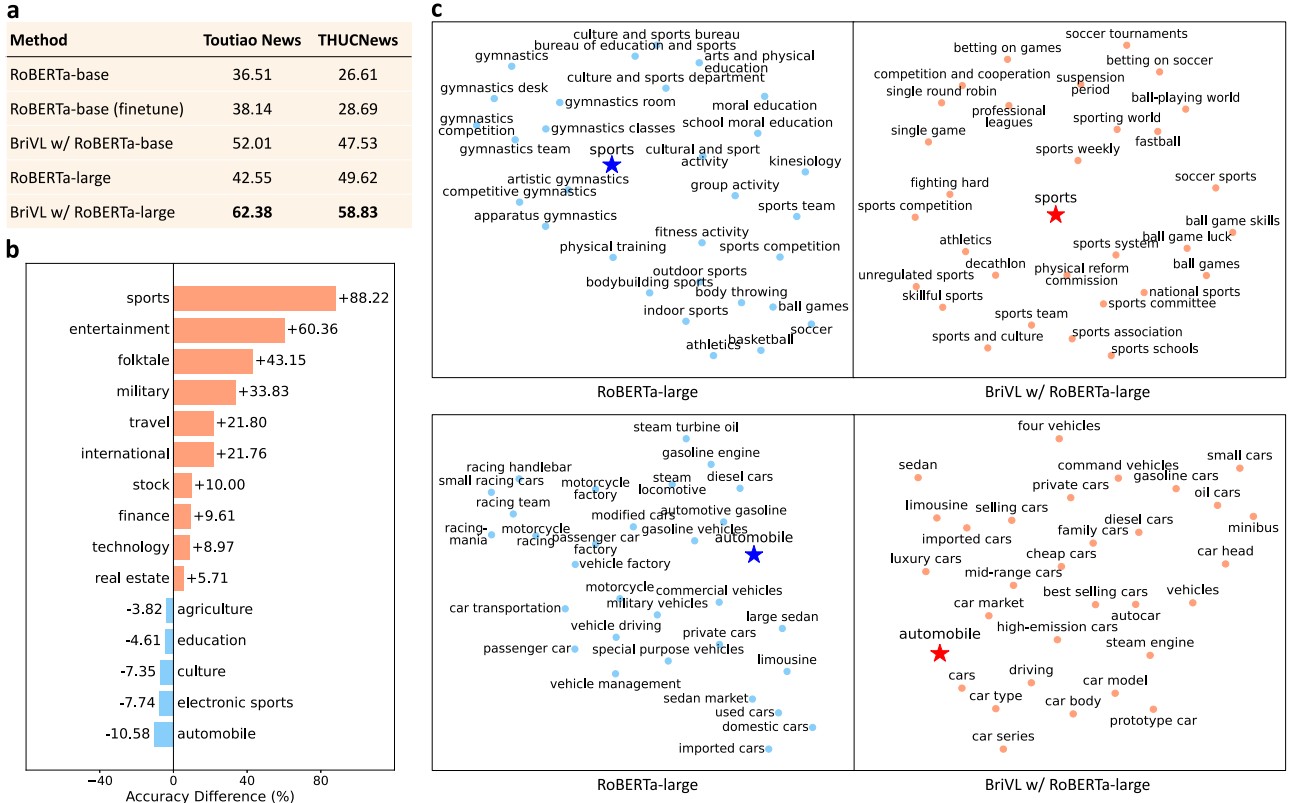

**Fig. 5 Zero-shot news classification results. a** Zero-shot news classification results (%) on two Chinese news datasets. **b** Zero-shot accuracy gain/loss (%) of BriVL w/ RoBERTa-large comparing to RoBERTa-large on each category of Toutiao News. **c** Top-30 phrase retrieval results of "sports" (top) and "automobile" (bottom) using RoBERTa-large and BriVL w/ RoBERTa-large, respectively. The candidate phrase list is obtained from Jieba, which consists of 347,728 Chinese phrases. We translate the results into English for presentation clarity. Highest results in (**a**) are highlighted in bold.

Furthermore, we deploy the aforementioned network visualization technique to clarify the visual responses of our BriVL to remote sensing related concepts. Concretely, we select one class "baseball field", and add the prompt "viewed from above" to the class name as the text input. The imagined visual content of our BriVL is shown in Fig. 4c along with one example of this class. We can see that remote sensing scenes are very different from traditional photos, mainly in the perspective of cameras. Despite this, we can observe from BriVL's imagination that there is a small sector-shaped area (marked with red lines) in "baseball field viewed from above". This provides direct explanation to the impressive performance of our BriVL on remote sensing scene classification. In addition, we search the keyword "baseball field" in our pre-training dataset WSCD and find that most of the related images are taken in a normal camera perspective. Given that there is hardly any remote sensing data in our WSCD, this finding suggests that BriVL has somehow learned to generalize transformation of perspectives to unseen domains during pre-training. This again shows the strong imagination ability and even hints of common sense reasoning ability of our BriVL.

**News classification.** To demonstrate how large-scale multimodal learning can benefit single-modal skills and also improve the imagination ability on single-modal tasks, we conduct zero-shot experiments on two Chinese news classification datasets. The first dataset is Toutiao News[40], which has 15 classes and a total of around 380K samples. The second dataset is THUCNews[41], which has 14 classes and around 840K samples in total. Since the contents in these two datasets are all texts, we only need the text encoder of our BriVL. Concretely, we first obtain class

embeddings by inputting class names into the text encoder. Further, for each piece of news, we only use its title to obtain its embedding via the text encoder. Finally, we compute the cosine similarities between each title embedding and class embeddings to make predictions.

The following methods are chosen for comparison: (1) RoBERTa-base:[42] it is an off-the-shelf Chinese language model pre-trained by the original authors on a large Chinese dataset with a total of 5.4B words. (2) RoBERTa-base (finetune): we finetune the pre-trained RoBERTa-base on a subset of our WSCD dataset (i.e., only the text data of 22M image-text pairs is used). (3) BriVL w/ RoBERTa-base: it is a small version of our standard BriVL as we reduce the CNN from EfficientNet-B7[43] to EfficientNet-B5 and also the text backbone from RoBERTa-large to RoBERTa-base. We pre-train this small version with the aforementioned 22M image-text pairs. (4) RoBERTa-large: it is the larger version of RoBERTa-base and is also pre-trained by the original authors. Its pre-training data is the same as that of RoBERTa-base. (5) BriVL w/ RoBERTa-large: our standard BriVL pre-trained on the whole WSCD.

The zero-shot classification results on Toutiao News and THUCNews are shown in Fig. 5a. It can be seen that: (1) The results of RoBERTa-base are lower than those of RoBERTa-large, which is expected since the latter has more parameters and a larger model capacity. (2) On both datasets, RoBERTa-base (finetune) has limited performance gains over RoBERTa-base, while BriVL w/ RoBERTa-base outperforms RoBERTa-base by large margins. This clearly indicates the advantage of cross-modal learning over single-modal learning, given that the finetuning data of RoBERTa-base (finetune) and BriVL w/ RoBERTa-base is both from the 22M subset of WSCD. (3) When it comes to

**a**

| Method | Fix BN | # Unfixed Blocks | Image-to-Text Retrieval | | | Text-to-Image Retrieval | | | Recall@SUM |
|---|---|---|---|---|---|---|---|---|---|
| | | | Recall@1 | Recall@5 | Recall@10 | Recall@1 | Recall@5 | Recall@10 | |
| BriVL (direct training) | no | 4 | 36.03 | 59.48 | 69.71 | 28.66 | 54.33 | 65.26 | 317.47 |
| BriVL (pre-train & finetune) | no | 2 | 43.05 | 65.11 | 74.57 | 32.49 | 57.53 | 67.99 | 342.74 |
| BriVL (pre-train & finetune) | no | 4 | 44.49 | 67.14 | 75.63 | 33.67 | 58.76 | 68.80 | 352.49 |
| BriVL (pre-train & finetune) | yes | 4 | **45.61** | **68.01** | **76.31** | **34.06** | **58.86** | **69.09** | **355.94** |

**b**

| Method | Fix BN | # Unfixed Blocks | Question Type | | | | | | Overall |
|---|---|---|---|---|---|---|---|---|---|
| | | | What | Where | When | Who | Why | How | |
| BriVL (direct training) | no | 4 | 70.51 | 71.99 | 81.88 | 77.05 | 78.36 | 68.62 | 72.16 |
| BriVL (pre-train & finetune) | no | 2 | **79.89** | **81.71** | 87.78 | **84.48** | 82.66 | **76.31** | **80.67** |
| BriVL (pre-train & finetune) | no | 4 | 79.41 | 81.66 | 87.31 | 84.46 | **83.11** | 74.44 | 80.16 |
| BriVL (pre-train & finetune) | yes | 4 | 77.79 | 80.50 | **87.99** | 84.44 | 82.46 | 72.87 | 78.96 |

**c**

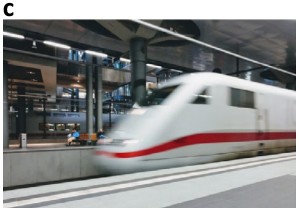 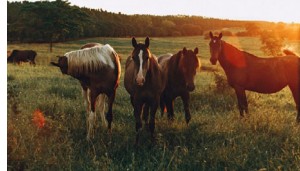 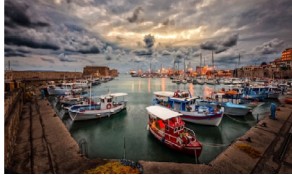 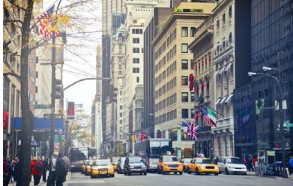

Why is the train blurry?
A. Moving fast.
B. Bad weather.
C. It's raining.
D. It's nighttime.
BriVL (direct training): C
BriVL (pre-train & finetune): A

Where was this taken?
A. In a zoo.
B. In a field.
C. On the street.
D. At the park.
BriVL (direct training): A
BriVL (pre-train & finetune): B

What are the boats doing?
A. Floating.
B. Sailing.
C. Getting cleaned.
D. Not moving.
BriVL (direct training): A
BriVL (pre-train & finetune): D

Why is the traffic stopped?
A. Car accident.
B. Traffic.
C. Red light.
D. Parade.
BriVL (direct training): B
BriVL (pre-train & finetune): C

**Fig. 6 Cross-modal retrieval and visual question answering (VQA) results. a** Cross-modal retrieval results (%) on the Chinese dataset AIC-ICC. **b** VQA results on Visual7W. Overall accuracies (%) along with results on each question type are reported. The dataset is translated into Chinese. **c** VQA examples of our BriVL model regarding whether it is pre-trained to validate the strong imagination ability of our pre-trained BriVL. Highest results in (**a**) and (**b**) are highlighted in bold.

RoBERTa-large, our BriVL w/ RoBERTa-large also leads to much better results than RoBERTa-large.

Moreover, in Fig. 5b, we present the performance gain/loss of our BriVL w/ RoBERTa-large comparing to RoBERTa-large on each category of Toutiao News. We can observe that the performance of BriVL decreases only on 5 categories but increases on the other 10, validating that the single-modal imagination/association ability can be improved by multimodal learning. Further, in Fig. 5c, we show top-30 phrase retrieval results of the category names "sports" and "automobile" using these two models to take a closer look. Concretely, we use a Chinese phrase list from Jieba[44] as the candidate list, which contains 347,728 phrases. Then we obtain the text embeddings of all candidates using RoBERTa-large and BriVL w/ RoBERTa-large, respectively. For each model and each category name, we compute the category name embedding and retrieve top-30 phrases by comparing it with all candidate embeddings using the cosine similarity. Finally, we visualize the results with the UMAP algorithm[45]. For "sports", we can see that our BriVL relates it to phrases with a higher variety than RoBERTa-large does. However, for "automobile", the retrieved top-30 phrases of our BriVL are more monotonous.

**Cross-modal retrieval**. Here we conduct experiments on the cross-modal retrieval downstream task, which is exactly what we train our BriVL to do. Since our BriVL is pre-trained with Chinese data, we choose the only available multimodal Chinese dataset AIC-ICC[46] for performance evaluation. AIC-ICC is originally designed for image captioning, which was first released in

AI Challenger 2017, a competition organized by Sinovation Ventures, Sogou, and Toutiao (ByteDance). The training set of AIC-ICC has 300K images and the validation set has 30K images. Each image has 5 Chinese captions. Since the test set is not released, we take the first 10K images along with their corresponding 50K pieces of texts from the validation set for testing.

The cross-modal retrieval results on AIC-ICC are shown in the table of Fig. 6a. The method "BriVL (direct training)" means that we directly train a randomly-initialized BriVL model on the training set of AIC-ICC rather than using the pre-trained BriVL. Moreover, the results of three "BriVL (pre-train & finetune)" variations are all obtained by finetuning our pre-trained BriVL on the training set of AIC-ICC with different finetuning strategies. We only consider two finetuning factors: whether to fix all the batch normalization (BN) layers in the CNN (i.e., EfficientNet-B7), and how many blocks should be unfixed in the CNN. We perform both image-to-text and text-to-image retrieval for evaluation, and report the results with Recall@k ($k = 1, 5, 10$) as well as Recall@SUM (i.e., the summation of six Recall@k metrics).

We can observe from the table of Fig. 6a that image-to-text retrieval results are generally higher than text-to-image ones, which is expected because like us humans, describing a given image is easier than imagining a picture from a sentence. We can also see that three "BriVL (pre-train & finetune)" variations achieve far better results than "BriVL (direct training)" for all evaluation metrics, indicating the usefulness of large-scale multimodal pre-training. In addition, using pre-trained models like our BriVL is more beneficial to image-to-text retrieval than to text-to-image retrieval, which may be due to the fact that image-to-text retrieval is an easier task. From the performance of three

"BriVL (pre-train & finetune)" variations, we find that different finetuning strategies do affect the final results, which should be kept in mind when we finetune pre-trained models for different downstream tasks.

**Visual question answering**. We consider another multimodal downstream task called visual question answering (VQA)[47] to further validate the strong imagination ability of our pre-trained BriVL on the Visual7W dataset[48]. Visual7W has 47.3K images from MSCOCO[49] and each image comes with a question and four answer candidates, where only one is the correct answer. The whole dataset can be divided into "Telling" questions and "Pointing" ones. Since "Pointing" questions rely on the bounding boxes of objects in images, we only conduct experiments on the "Telling" part, which can be further divided into six question types: "What", "Where", "When", "Who", "Why", and "How". We randomly make the training and test splits with 70% and 30%, respectively. Since Visual7W is an English dataset, we translate all of the questions and answer candidates into Chinese.

In the table of Fig. 6b, we report the overall accuracies on the test set, as well as the results on each question type. Similar to the situation on the cross-modal retrieval task, three "BriVL (pre-train & finetune)" variations achieve much better results than "BriVL (direct training)" for all question types, again indicating the usefulness of large-scale pre-training on downstream tasks. We also notice that the best finetuning strategy for cross-modal retrieval (i.e., fixing BN and keeping 4 blocks of the CNN unfixed) is no longer the best for VQA. In addition, although the strategy of not fixing BN and keeping 2 blocks unfixed obtains the best overall result, it does not achieve the best for all question types. This is expected as different tasks require different finetuning strategies.

Furthermore, we present four VQA examples in Fig. 6c. From these examples, we see our pre-trained BriVL clearly showing the strong imagination ability and even hints of common sense as it knows that the train in the picture looks blurry because it is moving fast, the picture of horses was taken in a field rather than in a zoo, the boats being tied to the dock are simply not moving instead of floating, and the traffic is stopped because of the red light instead of traffic jam. We believe that this is achieved by pre-training with our weak semantic correlation data: the texts are not detailed descriptions of their corresponding images, and thus our BriVL has to figure out the complicated connections hidden among this weak correlation during pre-training. With large pre-training data as much as 650 million, our BriVL finally succeeds in acquiring the ability of reasonably and logically imagining/ associating, and also manages to learn some common sense.

## Discussion

We have developed a large-scale multimodal foundation model called BriVL, which is efficiently trained on weak semantic correlation dataset (WSCD) consisting of 650 million image-text pairs. We have identified the direct evidence of the aligned image-text embedding space by neural network visualizations and text-to-image generation. In addition, we have visually revealed how a multimodal foundation model understands language and how it makes imagination or association about words and sentences. Moreover, extensive experiments on other downstream tasks show the cross-domain learning/transfer ability of our BriVL and the advantage of multimodal learning over single-modal learning. Particularly, our BriVL appears to acquire abilities in imagination and reasoning. Last but not least, we believe that all of these advantages are mainly due to the weak semantic correlation assumption followed by our BriVL. That is, by effectively fusing the complex human emotions and thoughts from those weakly

correlated image-text pairs, our BriVL is made more cognitive and general (i.e., much closer to AGI).

We believe that the solid step we take towards AGI would have a broad impact not only on the AI development community but also on a wide range of AI+ fields. For the AI research field itself, based on our GPU-resource-saving multimodal pre-training framework, researchers could easily extend our BriVL to a larger capacity with more modalities, leading to more general foundation models. Moreover, with the help of large-scale multimodal foundation models, it would also be much easier for researchers to explore novel tasks (especially those without abundant human-annotated samples). For AI+ fields, such foundation models could be quickly adapted to specific working context or environment, thanks to their strong generalization abilities. For example, in healthcare, multimodal foundation models could make full use of case data in multi-modality (e.g., computed tomography data, and blood routine examination data) to improve the diagnosing accuracy. Moreover, in neuroscience, multimodal foundation models could even help find out the mechanism of how multimodal data connect and fuse since artificial neural networks are much simpler to examine than real neural systems in human brains.

Nevertheless, multimodal foundation models still face potential risks and challenges. Since the performance of foundation models is based on the data that they are pre-trained on, it is likely that the models learn prejudices and stereotypes about certain issues, which should be carefully handled before model training and monitored/addressed in downstream applications. Moreover, as foundation models master more and more skills, creators of these models should be aware of model misuse by ill-intentioned people (e.g., manipulating or generating fake contents), which would have a negative influence on the society. In addition, on the evolution challenges of foundation models academically, it is of grand challenge for (1) developing model-interpretability tools deeper into the foundation models, (2) constructing huge pre-training datasets with more modalities, as well as (3) applying foundation models to various downstream tasks with more effective adaptation/finetuning techniques.

Our understanding of what BriVL (or any large-scale multi-modal foundation model) has learned and what it is capable of has only just started. There is still much room for further study to better understand the foundation model and develop more novel use cases. For instance, since the image can be regarded as a universally-understood "language", soliciting an even larger dataset containing multiple languages could result in a language translation model obtained as a by-product of multimodal pre-training. Moreover, additional modalities (e.g., videos and audios) can be also explored to pre-train a more intelligent model, taking us even closer to AGI.

## Methods

**Architecture overview**. The notion of pre-training a large-scale machine learning model and then using it for downstream tasks first appeared in natural language processing (NLP). As shown in Supplementary Note Fig. S1c, large-scale NLP models like GPT[50], BERT[51], and their variants, take Transformers[52] as text encoders to encode input texts into text embeddings, and then design pre-training objectives on top of these embeddings (e.g., generative loss and masked language modeling loss). In computer vision (CV), large-scale pre-training also becomes popular. Models like BiT[53] and ViT[54] use convolutional neural networks (CNNs) or Transformers as image encoders to obtain embeddings of input images. Similarly, pre-training losses are computed using the image embeddings (e.g., image classification loss and masked image patch prediction loss). However, these models are single-modal and thus only benefit downstream tasks in one modality. For multimodal (e.g., image, text, and audio) pre-training[12–18,32], existing works can be divided into two groups according to their network architectures: single-tower models (e.g., UNITER[17], OSCAR[12], and M6[18]) and two-tower ones (e.g., CLIP[13] and ALIGN[32]). Our BriVL can also be categorized into the two-tower models since we use separate image and text encoders. But note that we actually adopt two additional momentum encoders to help with the pre-training process (i.e., to

dynamically maintain negative sample queues across training batches), resulting in a four-tower pre-training architecture.

The pre-training goal of our BriVL is to learn two encoders that can embed image and text inputs into the same semantic space for effective image-text retrieval. To enforce the image and text encoders to learn better representations in the same embedding space, we introduce cross-modal contrastive learning with the InfoNCE loss[23] into our BriVL. Specifically, our learning objective is to find the corresponding image embedding from a batch of them for a given text embedding and vice versa. By maximizing the cosine similarity of the image and text embeddings for each ground-truth pair while minimizing the cosine similarities of the embeddings from negative pairs, we jointly train the image and text encoders to learn an aligned cross-modal embedding space.

Formally, for the image-text retrieval task, we denote the training set as $\mathcal{D} = \{(x_i^{(i)}, x_i^{(t)})|i = 1, \cdots, N\}$, where $(x_i^{(i)}, x_i^{(t)})$ is a matched image-text pair, and $N$ is the size of $\mathcal{D}$. Our BriVL leverages contrastive learning by applying MoCo[29] into the cross-modal scenario, as illustrated in Supplementary Note Fig. S1a. Each image $x_i^{(i)}$ (or each text $x_i^{(t)}$) is encoded by the image encoder $f^{(i)}$ (or the text encoder $f^{(t)}$) to obtain its $d$-dimensional embedding $\mathbf{z}_i^{(i)}$ (or $\mathbf{z}_i^{(t)}$). The image encoder (see Supplementary Note Fig. S1b) contains a CNN backbone, a successive self-attention (SA) block, and a multi-layer perceptron (MLP). A sequence of patch embeddings are first obtained by applying multi-scale patch pooling (MSPP) to the feature map from CNN. They are then fused/encoded by the SA block. The text encoder, on the other hand, is stacked by several SA blocks such as BERT[51] and RoBERTa[4]. A two-layer MLP block with a ReLU[55] activation layer is used for mapping each encoder's representation into the joint cross-modal embedding space. The parameters of $f^{(i)}$ and $f^{(t)}$ are denoted as $\theta^{(i)}$ and $\theta^{(t)}$, respectively.

**Image encoder**. To obtain better performance in the image-text retrieval task, most previous methods[19–21,56] utilize a bottom-up attention mechanism[57] with object features extracted by the Faster R-CNN detector[58]. However, extracting region/object features with a heavy detector is computationally expensive, e.g., a Faster R-CNN detector typically costs 0.064s (15.6 fps) to extract fine-grained region information from an image of moderate size. Meanwhile, the image-text retrieval would be inevitably limited by the detector's performance, which is not adaptable to real-world applications. In this paper, we thus introduce a simple yet effective module named Multi-Scale Patch Pooling (MSPP) to address this problem.

For each input image $x^{(i)}$, we first split it into multiple patches at different scales and record the patch coordinates. In all experiments, we take two scales as $1 \times 1$ and $6 \times 6$, resulting in a total of 37 patches. Next, we project each set of patch coordinates onto the feature map that is obtained by the CNN backbone (e.g., EfficientNet[43]) and generate a sequence of 37 region feature maps. Finally, we apply average pooling to each region feature map and obtain a sequence of patch features $\mathbf{S} \in \mathbb{R}^{c \times N_p}$, where each column corresponds to a patch, $N_p$ is the number of patches (i.e., $N_p = 37$ in this paper), and $c$ is the number of channels in the feature map.

To better capture the relationship of image patch features, we deploy a self-attention (SA) block containing multiple Transformer[52] encoder layers. Each Transformer encoder layer consists of a multi-head attention (MultiHeadAttn) layer and a feed forward network (FFN) layer:

$$\mathbf{S}' = \text{LayerNorm}(\mathbf{S} + \text{MultiHeadAttn}(\mathbf{S})) \tag{1}$$

$$\mathbf{S} = \text{LayerNorm}(\mathbf{S}' + \text{FFN}(\mathbf{S}')). \tag{2}$$

We then fuse the extracted patch features by applying an average pooling layer:

$$\mathbf{r}^{(i)} = \frac{1}{N_p} \sum_{j=1}^{N_p} \mathbf{S}_j \in \mathbb{R}^c, \tag{3}$$

where $\mathbf{S}_j$ is the $j$-th column of $\mathbf{S}$. A two-layer MLP block with a ReLU activation layer is adopted to project $\mathbf{r}^{(i)}$ to the joint cross-modal embedding space, resulting in the final $d$-dimensional image embedding $\mathbf{z}^{(i)} \in \mathbb{R}^d$.

**Text encoder**. Given a sentence $x^{(t)}$, we first tokenize it to obtain a sequence of tokens $\mathcal{T} = \{\mathbf{t}_j|j = 1, ..., l\}$, where $l$ denotes the length of the sentence (e.g., the number of words) and $\mathbf{t}_j$ denotes the $j$-th token of $\mathcal{T}$. A pre-trained Transformer encoder (e.g., RoBERTa[42]) is then used to map text tokens to a sequence of feature vectors (each feature vector corresponds to a word). Similarly, to better capture the relationship between words, we use the same self-attention mechanism as in the image encoder to extract the text representation $\mathbf{r}^{(t)}$. A two-layer MLP block with a ReLU activation layer is also used for mapping the text representation $\mathbf{r}^{(t)}$ to such joint cross-modal embedding space, resulting in the final $d$-dimensional text embedding $\mathbf{z}^{(t)} \in \mathbb{R}^d$.

**Contrastive loss**. The cross-modal contrastive loss in our BriVL is defined based on MoCo[29], which provides a mechanism of building dynamic sample queues for contrastive learning. Since the two negative queues used in our BriVL decouple the queue size from the mini-batch size, we can have a much larger negative sample size than the mini-batch size (thus GPU-resource-saving).

To maintain large queues of samples coming from different mini-batches and address the problem that sample features are extracted by encoders with very different parameters, we need two more smoothly updated encoders, that is, momentum encoders. The parameters $\theta_m^{(i)}$ (or $\theta_m^{(t)}$) of the momentum image encoder $f_m^{(i)}$ (or the momentum text encoder $f_m^{(t)}$) are updated in each training iteration with a momentum hyper-parameter $m$:

$$\theta_m^{(i)} = m \cdot \theta_m^{(i)} + (1 - m) \cdot \theta^{(i)}, \tag{4}$$

$$\theta_m^{(t)} = m \cdot \theta_m^{(t)} + (1 - m) \cdot \theta^{(t)}. \tag{5}$$

Further, we maintain two negative sample queues $\mathcal{Q}^{(i)}$ and $\mathcal{Q}^{(t)}$, which contain $N_q$ image negatives and $N_q$ text negatives for contrastive learning, respectively. In each pre-training iteration with the batch size $N_b$, all $N_b$ image negatives and $N_b$ text negatives are separately pushed into these two queues. Meanwhile, there are $N_b$ earliest samples being popped out of each queue. Concretely, at iteration $t$, the image and text negatives from the current data batch $(\mathcal{B}_t^{(i)}, \mathcal{B}_t^{(t)})$ are computed by respectively forwarding the momentum encoders $f_m^{(i)}$ and $f_m^{(t)}$:

$$\mathcal{N}_t^{(i)} = \left\{ f_m^{(i)}(x_i^{(i)})|x_i^{(i)} \in \mathcal{B}_t^{(i)} \right\}, \tag{6}$$

$$\mathcal{N}_t^{(t)} = \left\{ f_m^{(t)}(x_i^{(t)})|x_i^{(t)} \in \mathcal{B}_t^{(t)} \right\}, \tag{7}$$

where $|\mathcal{B}_t^{(i)}| = |\mathcal{B}_t^{(t)}| = N_b$. The obtained $\mathcal{N}_t^{(i)}$ and $\mathcal{N}_t^{(t)}$ are then pushed into $\mathcal{Q}^{(i)}$ and $\mathcal{Q}^{(t)}$, respectively. Note that although we generally call $\mathcal{N}_t^{(i)}$ (or $\mathcal{N}_t^{(t)}$) image negatives (or text negatives), there is still one sample being positive to each text (or image). Here, we denote the positive image sample (or text sample) for the $j$-th input text $x_j^{(t)}$ (or the $j$-th input image $x_j^{(i)}$) of the current mini-batch as:

$$\mathbf{p}_j^{(i)} = f_m^{(i)}(x_j^{(i)}) \in \mathcal{N}_t^{(i)}, \tag{8}$$

$$\mathbf{p}_j^{(t)} = f_m^{(t)}(x_j^{(t)}) \in \mathcal{N}_t^{(t)}. \tag{9}$$

With the two negative queues, the loss function in each training iteration is thus computed as follows. For each input image $x_i^{(i)}$, we define the contrastive loss between its image embedding $\mathbf{z}_i^{(i)}$ and all positive/negative texts in the queue $\mathcal{Q}^{(t)}$ as an InfoNCE loss[23]:

$$\mathcal{L}_{i2t} = -\frac{1}{N_b} \sum_i^{N_b} \log \frac{\exp\left(\mathbf{z}_i^{(i)} \cdot \mathbf{p}_i^{(t)}/\tau\right)}{\exp\left(\mathbf{z}_i^{(i)} \cdot \mathbf{p}_i^{(t)}/\tau\right) + \sum_{\mathbf{n}^{(t)}} \exp\left(\mathbf{z}_i^{(i)} \cdot \mathbf{n}^{(t)}/\tau\right)}, \tag{10}$$

where $\mathbf{n}^{(t)} \in \mathcal{Q}^{(t)} \setminus \{\mathbf{p}_i^{(t)}\}$ denotes a text negative for each image, $\tau$ is the temperature hyper-parameter, and the vector similarity is measured by dot product ($\cdot$). Similarly, for each input text $x_i^{(t)}$, the InfoNCE loss is given by:

$$\mathcal{L}_{t2i} = -\frac{1}{N_b} \sum_i^{N_b} \log \frac{\exp\left(\mathbf{z}_i^{(t)} \cdot \mathbf{p}_i^{(i)}/\tau\right)}{\exp\left(\mathbf{z}_i^{(t)} \cdot \mathbf{p}_i^{(i)}/\tau\right) + \sum_{\mathbf{n}^{(i)}} \exp\left(\mathbf{z}_i^{(t)} \cdot \mathbf{n}^{(i)}/\tau\right)}, \tag{11}$$

where $\mathbf{n}^{(i)} \in \mathcal{Q}^{(i)} \setminus \{\mathbf{p}_i^{(i)}\}$ denotes an image negative for each text. The total loss function for pre-training our BriVL is then defined as:

$$\mathcal{L}_{total} = \mathcal{L}_{i2t} + \mathcal{L}_{t2i}. \tag{12}$$

In the test/evaluation stage, given each query image (or text), the cross-modal retrieval results are obtained simply by the dot product defined over the outputs of the text (or image) encoder.

**Implementation details**. Over the input images, we adopt random graying and random color jittering for data augmentation. All images are resized to $600 \times 600$ pixels. We adopt EfficientNet-B7[43] as the CNN backbone in the image encoder and RoBERTa-Large[42] as the basis Transformer in the text encoder. For both image and text encoders, the self-attention block consists of 4 Transformer encoder layers and the MLP block has two fully-connected layers with a ReLU activation layer. The final embedding size of the joint cross-modal space is 2,560. We select the hyper-parameters heuristically for pre-training our BriVL model due to the computational constraint: the temperature hyper-parameter $\tau = 0.07$, momentum $m = 0.99$, and the queue size $N_q = 13,440$. We adopt the Adam optimizer[59], with the weight decay 1e-5 and the learning rate 1e-4. We use a mini-batch size of 192 for each of the 14 machines (each machine has 8 NVIDIA A100 GPUs), resulting in a total batch size of 2688 (far smaller than $N_q$). The resource-saving advantages of such batch setting are shown by the ablation study results in Supplementary Note Fig. S2. We also deploy the latest distributed-training framework DeepSpeed[26] to accelerate the pre-training process and save the GPU memories. With 112 NVIDIA A100 GPUs in total, it takes about 10 days to pre-train our BriVL model over our WSCD of 650 million image-text pairs.

**Differences from CLIP/ALIGN**. We have stated two main differences between our BriVL and CLIP/ALIGN in the Introduction section. Below we give more detailed

differences technically. (1) We adopt a four-tower network architecture (see Supplementary Note Fig. S1a) for pre-training. By extending the original single-modal contrastive learning (CL) algorithm MoCo[29], we introduce momentum encoders and negative sample queues for multimodal pre-training in a more GPU-resource-saving way. In contrast, both CLIP and ALIGN employ the standard two-tower architecture, which requires large batch size (thus enough negative samples) to be effective, taking up a mass of GPU memories. (2) We additionally devise a multi-scale patch pooling (MSPP) module (see Supplementary Note Fig. S1b) to capture fine-grained image region representations without using object detectors. While CLIP and ALIGN only consider global-level image embeddings, which impedes their ability to learn fine-grained/local image features.

**Formalization of neural network visualization**. Neural network visualization is developed to directly show the visual response/imagination of BriVL w.r.t. the semantic input. Formally, given the pre-trained image and text encoders $f^{(i)}$ and $f^{(t)}$ of BriVL, we first input a piece of text $x^{(t)}$ and obtain its text embedding $z^{(t)} = f^{(t)}(x^{(t)}) \in \mathbb{R}^d$. In the mean time, we randomly initialize a noisy image $x^{(i)} \in \mathbb{R}^{600 \times 600 \times 3}$, which contains all the learnable parameters throughout the entire visualization process. Further, we obtain the image embedding $z^{(i)} = f^{(i)}(x^{(i)}) \in \mathbb{R}^d$ and define the learning objective by matching the two embeddings:

$$\mathcal{L}_{vis} = -\cos(z^{(i)}, z^{(t)}), \tag{13}$$

where $\cos(\cdot, \cdot)$ computes the cosine similarity between two vectors. With the resultant gradients, we are able to update the input image $x^{(i)}$ by back-propagation. After repeating the above updating step with multiple iterations, we finally obtain an image $x^{(i)}$, which can be regarded as BriVL's response/imagination about the input text. The algorithm for neural network visualization is summarized in Algorithm 1.

**Algorithm 1**. Neural Network Visualization
**Input:** The pre-trained image and text encoder $f^{(i)}$ and $f^{(t)}$ of our BriVL
　　　　A piece of text $x^{(t)}$
　　　　A randomly initialized image $x^{(i)}$
　　　　A learning rate parameter $\lambda$
**Output:** The updated input image
1: Obtain the text embedding $z^{(t)} = f^{(t)}(x^{(t)})$;
2: **for all** iteration $= 1, 2, \cdots$, MaxIteration **do**
3:　　　Obtain the image embedding $z^{(i)} = f^{(i)}(x^{(i)})$;
4:　　　Compute $\mathcal{L}_{vis}$ with Eq. (13);
5:　　　Compute the gradients $\nabla_{x^{(i)}} \mathcal{L}_{vis}$;
6:　　　Update $x^{(i)}$ using gradient descent with $\lambda$;
7: **end for**
8: **return** the updated input image.

**Formalization of text-to-image generation**. To make BriVL's response/imagination on input texts better understood, we further adopt VQGAN[34] to help generate more photo-realistic images. The reason of utilizing VQGAN instead of other GANs[60] is as follows. Although classic GANs are able to generate high quality images under specific domains (e.g., natural sceneries or human faces), they tend to fail when complex scenarios are involved. In contrast, VQGAN alleviates this problem and performs better under complex scenarios by combining VQVAE[61] and GAN. For our text-to-image generation, we only need a codebook $\mathcal{C}$ and a CNN generator $g$ of the VQGAN pre-trained on ILSVRC-2012[35]. The pre-trained codebook $\mathcal{C} = \{c_k \in \mathbb{R}^{d_c} | k = 1, 2, \cdots, N_c\}$ is a collection of tokens/codes, where $d_c$ is the dimension of each code and $N_c$ is the number of codes in the codebook ($d_c = 256$ and $N_c = 1,024$ in our case). The pre-trained CNN generator $g$ takes a spatial collection of codebook entries $U \in \mathbb{R}^{h \times w \times d_c}$ as input to generate an image ($U$ can also be regarded as a sequence of $hw$ codes, $h = w = 16$ in our case), where each element $u_{ij} \in \mathbb{R}^{d_c}$ ($i = 1, 2, \cdots, h$ and $j = 1, 2, \cdots, w$) must come from the codebook $\mathcal{C}$ (i.e., $u_{ij} \in \mathcal{C}$). With the pre-trained image and text encoders $f^{(i)}$ and $f^{(t)}$ of BriVL, we first input a piece of text $x^{(t)}$ and obtain its text embedding $z^{(t)} = f^{(t)}(x^{(t)}) \in \mathbb{R}^d$. Meanwhile, we randomly initialize an input code collection $U$, which is the only parameter matrix to be learned. Afterwards, we generate an image from the generator $x^{(i)} = g(U)$ and further obtain its image embedding $z^{(i)} = f^{(i)}(x^{(i)}) \in \mathbb{R}^d$. The learning objective is to maximize the similarity between two embeddings:

$$\mathcal{L}_{t2i} = -\cos(z^{(i)}, z^{(t)}). \tag{14}$$

After updating the input $U$ and obtaining $U'$ by back-propagation, we need to perform an element-wise quantization of each spatial code $u'_{ij} \in \mathbb{R}^{d_c}$ in $U'$ onto its closest codebook entry $c_k$:

$$u_{ij} = \arg\min_{c_k \in \mathcal{C}} \|u'_{ij} - c_k\|. \tag{15}$$

By repeating the above updating step with multiple iterations, we finally obtain an image $x^{(i)}$ generated with the updated $U$. The algorithm for text-to-image generation is summarized in Algorithm 2.

**Algorithm 2**. Text-to-Image Generation
**Input:** The pre-trained image and text encoder $f^{(i)}$ and $f^{(t)}$ of our BriVL
　　　　The codebook $\mathcal{C}$ and the CNN generator $g$ of the pre-trained VQGAN
　　　　A piece of text $x^{(t)}$
　　　　A randomly initialized collection of codebook entries $U$
　　　　A learning rate parameter $\lambda$
**Output:** The image generated with the updated $U$
1: Obtain the text embedding $z^{(t)} = f^{(t)}(x^{(t)})$;
2: **for all** iteration $= 1, 2, \cdots$, MaxIteration **do**
3:　　　Generate an image $x^{(i)} = g(U)$;
4:　　　Obtain the image embedding $z^{(i)} = f^{(i)}(x^{(i)})$;
5:　　　Compute $\mathcal{L}_{t2i}$ with Eq. (14);
6:　　　Compute the gradients $\nabla_U \mathcal{L}_{t2i}$;
7:　　　Obtain $U'$ by updating $U$ using gradient descent with $\lambda$;
8:　　　Obtain $U$ by performing element-wise quantization on $U'$ with Eq. (15);
9: **end for**
10: **return** the image generated with the updated $U$.

**Neural network visualization vs. text-to-image generation**. The intrinsic difference between neural network visualization and text-to-image generation lies in that they produce images following different data distributions. Not utilizing extra modules or data, neural network visualization exhibits BriVL's primitive visual understanding of a given piece of text. However, the VQGAN[34] used for text-to-image generation is pre-trained on ILSVRC-2012[35] (i.e., the classic ImageNet dataset), which generates images following the data distribution of ImageNet and thus being more photo-realistic. Due to such an intrinsic difference, we present the visualization results of these two tasks for different purposes in this paper. Specifically, neural network visualization allows us to see what exactly a pre-trained multi-modal foundation model imagines about semantic concepts and sentences, while text-to-image generation is used to generate images matched with given texts in a more human-friendly way.

## Data availability

The availability of datasets used in this study is detailed as follows: (1). Two remote sensing scene classification datasets: UC Merced Land-Use (UCM, http://weegee.vision.ucmerced.edu/datasets/landuse.html) and AID (https://captain-whu.github.io/AID/). (2). Two news classification datasets: THUCNews (http://thuctc.thunlp.org/) and Toutiao News (https://github.com/aceimnorstuvwxz/toutiao-text-classfication-dataset). (3). The Chinese cross-modal retrieval dataset AIC-ICC is available at https://github.com/neilfei/brivl-nmi. (4). The VQA dataset Visual7W is available at http://ai.stanford.edu/yukez/visual7w/. (5). The dataset used for pre-training our model is available at https://resource.wudaoai.cn/home. Please note that the available datasets (1) – (4) are sufficient for finetuning our pre-trained BriVL model in order to interpret, verify and extend our research.

## Code availability

The pre-trained BriVL and its inference code are available at https://github.com/neilfei/brivl-nmi under Creative Commons Attribution-Non Commercial-No Derivatives 4.0 International Licence (CC BY-NC-ND).

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

## Acknowledgements

Z.L. acknowledges National Natural Science Foundation of China (61976220). J.R.W. acknowledges National Natural Science Foundation of China (61832017), Beijing Outstanding Young Scientist Program (BJJWZYJH012019100020098), and Large-Scale Pre-Training Program 468 of Beijing Academy of Artificial Intelligence (BAAI). N.F. acknowledges the Outstanding Innovative Talents Cultivation Funded Programs 2021 of Renmin Univertity of China. We acknowledge the WenLan Data Group for helping us collect the pre-training dataset.

## Author contributions

Z.L. contributed the original idea, model design, and experimental analysis. Z.L. and N.F. wrote the majority of the manuscript. N.F., Y.G. and Y.H. contributed the source code. The experiments were conducted by N.F., G.Y., J.W., H.L. and Y.G. The review and editing of the manuscript were carried out by H.S., T.X., X.G., R.S. and J.R.W. The entire project was supervised by J.R.W.

## Competing interests

The authors declare no competing interests.

## Additional information



