## [Peer Review File · Nature Communications]

Towards artificial general intelligence via a multimodal foundation modelREVIEWER COMMENTS

Reviewer #1 (Remarks to the Author):

Summary

This paper presents a very early exploration of how to realize task-independent generalized AI based on a multi-modal pre-training protocol. Specifically, they provide a series of key improvements from multiple dimensions, including data collection, model construction, and model interpretation.

Major Contribution

Data Construction: Existing multi-modal pre-training methods rely on the strong semantic correlation across text and image, which requires enormous human labor and a heavy computational burden. To address this issue, this paper constructs a novel pre-training dataset based on weak semantic correlation where only the global alignment between text and image is required. Such data could be obtained efficiently from the Internet without further human annotation.

Model Construction: To improve the pre-training efficiency, this paper proposed a two-tower architecture to perform contrastive learning. Moreover, the momentum mechanism is also introduced to reinforce the memory of the neural net and thus reduce the batch size.

Model Interpretation: On top of the pre-training model, this paper presents two methods to visualize the text information based on the given model. The imagined image could be learned automatically from either FeaVis or VQGAN.

Comments

Writing: This paper seems easy-to-read with a clear motivation and a clean organization of the contents.

Details: This paper provides a novel large-scale pre-training dataset, a novel pre-training method, and most interestingly, an intelligent way to visualization the imagination of the trained model. Overall, I believe the technical details therein are solid.

Significance: To the best of my knowledge, this paper conducts a very early trial on generalized AI on top of a multi-modal pre-training framework. Moreover, as its name implies, it is also the first time to present a visualization scheme of how multi-modal imagines when text information is given.

Noteworthy Results: It is fascinating to see that the weak semantic correlation simplifies the pre-training process and brings even stronger generalization ability. Evidence of such an argument could be found through the analysis in their almost exhaustive experimental results. Moreover, the results in Fig. 2 and Fig. 3 show that the novel pre-training model could leverage clear imaginations from the words/sentences given.

Relevance: Aiming to break the task-driven limitation, this paper focuses on the task-independent general AI framework. The pre-trained model is expected to perform well on various tasks. This is one of the most important open problems for the AI community. Hence, I think this paper could raise wide attention from the related researchers.

Reproducibility: This paper provides clear implementation details of their experiments and

the data collection process. Moreover, both the test datasets and the source code are available online.

Overall, I think this is a good paper with solid methodology and experiments, which would be significant for future studies toward general AI/Machine Learning.

I only have some minor concerns for the authors.

Minor Concerns

1) It seems that the description of the model visualization is a bit lengthy. Perhaps the authors might want to re-organize it into an algorithm. Moreover, the objective function to learn the imagined image should also be given.

2) The "Text-Image-Generation" section seems to have a similar problem. The authors may wish to provide a more compact and accurate expression in the form of an algorithm. Also, I think the authors should explain why adopting the VQGAN for visualization.

3) It seems that the title of this paper focuses on imagination or, namely, the interpretation of the model features/results. However, from my perspective, the primal goal of this paper is to improve the multi-modal pre-training method, and the interpretation is only one of its products. Hence, it would be better to make it more clear in the title.

Reviewer #2 (Remarks to the Author):

This paper introduces an innovative and transformative framework that masters key capacities (e.g., imagination) of AGI. In particular, a foundation model is established and pre-trained on a large set of weak semantic multi-modal data through contrastive self-supervised learning. The model interpretability and generalizability are clearly demonstrated via neural visualization, text-to-image generation, as well as several downstream tasks. In general, this is an excellent paper, well organized and written. The idea is novel and the numerical result is solid to support the conclusion. I trust that the proposed model has great potential to lead to major impact on various AI+ fields and will draw broad interest in the computational science community. Hence, I recommend the paper for publication in Nature Communications.

Here, I only have a couple of minor questions on the details of neural network visualization and text-to-image generation.

1. Although the authors had described the process of neural network visualization and text-to-image generation in the paper, it is not exactly clear how they were performed. Please provide more implementation details regarding these two tasks. It would be better to describe them formally or provide the source code.

2. If I understand correctly, the direct difference between neural network visualization and text-to-image generation is that the latter introduces an extra VQGAN module. However, considering that these two tasks are both performed with all neural network parameters frozen in an inversion way, what are the intrinsic differences between them?

Towards artificial general intelligence via a multimodal foundation model

point-by-point response

Nanyi Fei^{1,2,3}, Zhiwu Lu^{1,2}, Yizhao Gao^{1,2}, Guoxing Yang^{1,2}, Yuqi Huo^{2,3}, Jingyuan Wen^{1,2}, Haoyu Lu^{1,2}, Ruihua Song^{1,2}, Xin Gao⁴, Tao Xiang⁵, Hao Sun^{1,2,6} and Ji-Rong Wen^{1,2,3}

¹*Gaoling School of Artificial Intelligence, Renmin University of China, Beijing, China*

²*Beijing Key Laboratory of Big Data Management and Analysis Methods, Beijing, China*

³*School of Information, Renmin University of China, Beijing, China*

⁴*Computer, Electrical and Mathematical Sciences and Engineering Division, King Abdullah University of Science and Technology, Thuwal, Saudi Arabia*

⁵*Department of Electrical and Electronic Engineering, University of Surrey, Guildford, United Kingdom*

⁶*Department of Civil and Environmental Engineering, Massachusetts Institute of Technology, Cambridge, USA*

We are sincerely grateful to the editor and reviewers for their encouraging and constructive comments and suggestions. Our response is provided following the comments marked in blue color. Revisions have also been made in the revised manuscript where indicated (in red color).

Major Changes to Our Manuscript

Before giving our point-by-point responses to the comments made by the two reviewers, we first provide a quick summary of the major changes to our manuscript as follows:

- (1). The title of our manuscript has been changed to “Towards artificial general intelligence via a multimodal foundation model”.
- (2). Two paragraphs “Formalization of Neural Network Visualization” and “Formalization of Text-to-Image Generation” have been added in the Methods section, which formally describe the details of neural network visualization and text-to-image generation, respectively.
- (3). Two organized algorithms for neural network visualization and text-to-image generation have been added as Algorithm 1 and Algorithm 2, respectively.
- (4). A paragraph “Neural Network Visualization vs. Text-to-Image Generation” has been added in the Methods section to clarify the difference between neural network visualization and text-to-image generation.
- (5). The network architecture figure (originally as Extended Data Fig. 1 in the main manuscript) has been moved to the supplementary note, and we have added a section “Architecture Overview” in the supplementary note to reflect this change.

32 **Response to Reviewer #1**

33 Summary

34 This paper presents a very early exploration of how to realize task-independent generalized AI based on
35 a multi-modal pre-training protocol. Specifically, they provide a series of key improvements from multiple
36 dimensions, including data collection, model construction, and model interpretation.

37 Major Contribution

38 **Data Construction:** Existing multi-modal pre-training methods rely on the strong semantic correlation
39 across text and image, which requires enormous human labor and a heavy computational burden. To address
40 this issue, this paper constructs a novel pre-training dataset based on weak semantic correlation where only
41 the global alignment between text and image is required. Such data could be obtained efficiently from the
42 Internet without further human annotation.

43 **Model Construction:** To improve the pre-training efficiency, this paper proposed a two-tower architec-
44 ture to perform contrastive learning. Moreover, the momentum mechanism is also introduced to reinforce
45 the memory of the neural net and thus reduce the batch size.

46 **Model Interpretation:** On top of the pre-training model, this paper presents two methods to visualize
47 the text information based on the given model. The imagined image could be learned automatically from
48 either FeaVis or VQGAN.

49 Comments

50 **Writing:** This paper seems easy-to-read with a clear motivation and a clean organization of the contents.

51 **Details:** This paper provides a novel large-scale pre-training dataset, a novel pre-training method, and
52 most interestingly, an intelligent way to visualization the imagination of the trained model. Overall, I believe
53 the technical details therein are solid.

54 **Significance:** To the best of my knowledge, this paper conducts a very early trial on generalized AI on top
55 of a multi-modal pre-training framework. Moreover, as its name implies, it is also the first time to present
56 a visualization scheme of how multi-modal imagines when text information is given.

57 **Noteworthy Results:** It is fascinating to see that the weak semantic correlation simplifies the pre-training
58 process and brings even stronger generalization ability. Evidence of such an argument could be found through
59 the analysis in their almost exhaustive experimental results. Moreover, the results in Fig. 2 and Fig. 3 show
60 that the novel pre-training model could leverage clear imaginations from the words/sentences given.

61 **Relevance:** Aiming to break the task-driven limitation, this paper focuses on the task-independent general
62 AI framework. The pre-trained model is expected to perform well on various tasks. This is one of the most
63 important open problems for the AI community. Hence, I think this paper could raise wide attention from
64 the related researchers.

65 **Reproducibility:** This paper provides clear implementation details of their experiments and the data
66 collection process. Moreover, both the test datasets and the source code are available online.

67 Overall, I think this is a good paper with solid methodology and experiments, which would be significant
68 for future studies toward general AI/Machine Learning. I only have some minor concerns for the authors.

69 **RE:** We sincerely thank the reviewer for recognizing and highlighting the contributions and the novelty of
70 our work, along with positive and encouraging comments.
71

72 Minor Concerns

73 1. It seems that the description of the model visualization is a bit lengthy. Perhaps the authors might want
74 to re-organize it into an algorithm. Moreover, the objective function to learn the imagined image should
75 also be given.

76 **RE:** Thanks for this nice suggestion. We have added a paragraph “Formalization of Neural Network Vi-
77 sualization” in the Methods section (see Page 13), which formally describes the details of neural network
78 visualization (including its objective function). Moreover, we have also re-organized the process into an
79 algorithm as illustrated in Algorithm 1 (see Page 14).

80 2. The “Text-to-Image Generation” section seems to have a similar problem. The authors may wish to
81 provide a more compact and accurate expression in the form of an algorithm. Also, I think the authors
82 should explain why adopting the VQGAN for visualization.

83 **RE:** Thanks for this thoughtful comment. We have added another paragraph “Formalization of Text-to-
84 Image Generation” in the Methods section together with an organized algorithm (see Algorithm 2). Please
85 see Pages 13-14 in the revised manuscript. The reason of utilizing VQGAN instead of other GANs is as
86 follows. Although classic GANs are able to generate high quality images under specific domains (e.g., natural
87 sceneries or human faces), they tend to fail when complex scenarios are involved. In contrast, VQGAN can
88 alleviate this problem, i.e., it performs better under complex scenarios by combining VQVAE and GAN.
89 As the process of our text-to-image generation is very flexible, VQGAN can be replaced by other GANs
90 for generating images under their well-performing domains (e.g., adopting StyleGAN for generating human
91 faces). We have included this discussion in the “Formalization of Text-to-Image Generation” paragraph.

92 3. It seems that the title of this paper focuses on imagination or, namely, the interpretation of the model
93 features/results. However, from my perspective, the primal goal of this paper is to improve the multi-modal
94 pre-training method, and the interpretation is only one of its products. Hence, it would’ve been better to
95 make it more clear in the title of this paper.

96 **RE:** This is a great suggestion. After a careful consideration, we have changed the paper title to “Towards
97 artificial general intelligence via a multimodal foundation model”.

98 **Response to Reviewer #2**

99 This paper introduces an innovative and transformative framework that masters key capacities (e.g., imag-
100 ination) of AGI. In particular, a foundation model is established and pre-trained on a large set of weak
101 semantic multi-modal data through contrastive self-supervised learning. The model interpretability and
102 generalizability are clearly demonstrated via neural visualization, text-to-image generation, as well as sev-
103 eral downstream tasks. In general, this is an excellent paper, well organized and written. The idea is novel
104 and the numerical result is solid to support the conclusion. I trust that the proposed model has great
105 potential to lead to major impact on various AI+ fields and will draw broad interest in the computational
106 science community. Hence, I recommend the paper for publication in Nature Communications.

107 Here, I only have a couple of minor questions on the details of neural network visualization and text-to-image
108 generation.

109 **RE:** We sincerely thank the reviewer for the very positive and encouraging feedback.

110 1. Although the authors had described the process of neural network visualization and text-to-image gen-
111 eration in the paper, it is not exactly clear how they were performed. Please provide more implementation

112 details regarding these two tasks. It would be better to describe them formally or provide the source code.

113 **RE:** Thanks for this nice suggestion. We have added two paragraphs “Formalization of Neural Network
114 Visualization” and “Formalization of Text-to-Image Generation” in the Methods section (see Pages 13-14),
115 which formally describe the process of neural network visualization and text-to-image generation, respec-
116 tively. Moreover, we have also re-organized them into two algorithms (see Algorithm 1 and Algorithm
117 2). The source codes of neural network visualization and text-to-image generation will be added to our
118 repository at <https://github.com/neilfei/brivl-nmi> after the paper is accepted for publication.

119 2. If I understand correctly, the direct difference between neural network visualization and text-to-image
120 generation is that the latter introduces an extra VQGAN module. However, considering that these two tasks
121 are both performed with all neural network parameters frozen in an inversion way, what are the intrinsic
122 differences between them?

123 **RE:** Thanks for this question. The intrinsic difference between neural network visualization and text-to-
124 image generation lies in that they produce images following different data distributions. Not utilizing extra
125 modules or data, neural network visualization exhibits BriVL’s primitive visual understanding of a given
126 piece of text. However, the VQGAN used for text-to-image generation is pre-trained on ILSVRC-2012 (i.e.,
127 the classic ImageNet dataset), which generates images following the data distribution of ImageNet and thus
128 being more photo-realistic. Due to such intrinsic difference, we present the visualization results of these two
129 tasks for different purposes in this paper. Specifically, neural network visualization allows us to see what
130 exactly a pre-trained multi-modal foundation model imagines about semantic concepts and sentences, while
131 text-to-image generation is used to generate images matched with given texts in a more human-friendly way.
132 Note that an image outputted by neural network visualization may contain too much information and thus
133 its partial content may hardly be understood by human, but an image generated by VQGAN is more likely
134 to be understandable to human with the constraint of following the ImageNet data distribution. Hence,
135 we have added a paragraph “Neural Network Visualization vs. Text-to-Image Generation” in the Methods
136 section (see Page 14) to clarify this aspect.

REVIEWER COMMENTS

Reviewer #1 (Remarks to the Author):

In the current version, the authors have addressed all my concerns about the presentation and organization. Consequently, the quality of this paper is improved significantly. I would like to suggest accepting this paper, especially seeing that the other reviewer is also quite positive.

Reviewer #2 (Remarks to the Author):

The authors have addressed my comments. I am OK with publication.

Towards artificial general intelligence via a multimodal foundation model

point-by-point response

Nanyi Fei^{1,2,3}, Zhiwu Lu^{1,2}, Yizhao Gao^{1,2}, Guoxing Yang^{1,2}, Yuqi Huo^{2,3}, Jingyuan Wen^{1,2}, Haoyu Lu^{1,2}, Ruihua Song^{1,2}, Xin Gao⁴, Tao Xiang⁵, Hao Sun^{1,2} and Ji-Rong Wen^{1,2,3}

¹*Gaoling School of Artificial Intelligence, Renmin University of China, Beijing, China*

²*Beijing Key Laboratory of Big Data Management and Analysis Methods, Beijing, China*

³*School of Information, Renmin University of China, Beijing, China*

⁴*Computer, Electrical and Mathematical Sciences and Engineering Division, King Abdullah University of Science and Technology, Thuwal, Saudi Arabia*

⁵*Department of Electrical and Electronic Engineering, University of Surrey, Guildford, United Kingdom*

We are sincerely grateful to the reviewers for their positive feedback. Our response is provided following the editor’s comments marked in blue color. Revisions have also been made in the revised manuscript where indicated (in red color).

Major Changes to Our Manuscript

Before giving our point-by-point responses to the comments made by the editor and the two reviewers, we first provide a quick summary of the major changes to our manuscript as follows:

- (1). The cake image in Fig. 1b has been replaced by a similar image from the Pexels website¹.
- (2). Fig. 4d has been removed. The “baseball field” image in Fig. 4c has been replaced by a similar image from the Pexels website. The caption of Fig. 4 and the corresponding content of the manuscript have been revised.
- (3). The second VQA example in Fig. 6c has been replaced by an image (without humans) from the Pexels website, and the corresponding content of the manuscript has been revised.
- (4). The second and third rows of the original Fig. S2b have been removed. All images in Fig. S2b have then been replaced by those from the Pexels website (without identifiable humans). The caption of Fig. S2 and corresponding content of the manuscript has been revised.
- (5). All three image captioning examples in Fig. S3c have been replaced by images (without humans) from the Pexels website, and the corresponding content of the manuscript has been revised.
- (6). A new section “Image Sources” has been added in the supplementary note, where the sources of images/elements in all figure panels, when appropriate, have been listed in Table S1.

¹<https://www.pexels.com/>

30 Response to the Editor

31 1. Remove all images including humans that can be identified or replace with other images not including
32 humans (the replacement images are should comply with journal’s CC BY license).

33 **RE:** Thanks for this suggestion. We have removed or replaced all images including humans that can be
34 identified. The replaced images are all taken from the Pexels website (<https://www.pexels.com>), which
35 provides free stock photos and allows users to download for free use (see its license page “<https://www.pexels.com/license/>” for more information). We believe the replaced images all comply with the journal’s
36 CC BY license.
37

38 2. All images or elements need querying. Please provide a table as a related manuscript file with full
39 information for all images taken from open datasets (dataset URL, image identification number).

40 **RE:** Thanks for this suggestion. All the third-party images used in both the main manuscript and the
41 supplementary note are taken from the Pexels website. We have added a new section “Image Sources” in
42 the supplementary note, where the sources of images/elements in all figure panels, when appropriate, are
43 listed in Table S1.

44 3. Fig. 4d - remove panel ‘d’.

45 **RE:** Thanks. We have removed Fig. 4d.

46 4. Fig. 6c – replace second panel with another image not including humans (the replacement images should
47 comply with journal’s CC BY license).

48 **RE:** We have replaced the second panel in Fig. 6c with another image from Pexels, which now does not
49 include humans.

50 5. SI, Fig. S2 – remove second- and third-line examples, leave examples in the first line (cup of tea), and last
51 line (fire, excluding second panel from the left). The source of the cup of tea and fire needs to be confirmed.
52 Please provide information whether you have permission to reuse this image in a CC BY licensed paper.

53 **RE:** We have removed the second and third rows of the original Fig. S2b. All images in Fig. S2b (the tea
54 cup image and the five fire images without identifiable humans) have then been replaced by those from the
55 Pexels website.

56 6. SI, Fig. S3 – replace images with other examples not including humans, or make own photos with
57 individuals signing permission to publish (All images should be in agreement with the CC BY license. If the
58 images consist of humans, they need to sign a release form confirming their agreement to have the image
59 published in a CC BY licensed paper).

60 **RE:** We have replaced all three panels in Fig. S3c with images from Pexels, which do not include humans.

61 Response to Reviewer #1

62 In the current version, the authors have addressed all my concerns about the presentation and organization.
63 Consequently, the quality of this paper is improved significantly. I would like to suggest accepting this paper,
64 especially seeing that the other reviewer is also quite positive.

65 **RE:** We sincerely thank the reviewer for the very positive and encouraging feedback.

66 **Response to Reviewer #2**

67 The authors have addressed my comments. I am OK with publication.

68 **RE:** We sincerely thank the reviewer for the very positive and encouraging feedback.